# Validity and reliability of the diagnostic codes for hypochondriasis and dysmorphophobia in the Swedish National Patient Register: a retrospective chart review

Daniel Rautio ,[1,2] Alba Vilaplana-Pérez ,[1,2] Martina Gumpert,[1,2] Volen Z Ivanov,[1,2] Johanna Linde,[1] Susanna Österman,[1,2] Oskar Flygare ,[1,2] Josef Isung,[1,2] Kayoko Isomura,[1,2] Sonja Krig,[2] Eva Serlachius,[1,2] Jens Högström,[1,2] Christian Rück ,[1,2] David Mataix-Cols ,[1,2] Lorena Fernández de la Cruz [1,2]

[1]Centre for Psychiatry Research, Department of Clinical Neuroscience, Karolinska Institutet, Stockholm, Sweden
[2]Stockholm Health Care Services, Region Stockholm, Stockholm, Sweden

**Correspondence to**
Daniel Rautio;
daniel.rautio@ki.se

## ABSTRACT

**Objectives** In the International Classification of Diseases, Tenth Edition (ICD-10), hypochondriasis (illness anxiety disorder) and dysmorphophobia (body dysmorphic disorder) share the same diagnostic code (F45.2). However, the Swedish ICD-10 allows for these disorders to be coded separately (F45.2 and F45.2A, respectively), potentially offering unique opportunities for register-based research on these conditions. We assessed the validity and reliability of their ICD-10 codes in the Swedish National Patient Register (NPR).

**Design** Retrospective chart review.

**Methods** Six hundred individuals with a diagnosis of hypochondriasis or dysmorphophobia (300 each) were randomly selected from the NPR. Their medical files were requested from the corresponding clinics, located anywhere in Sweden. Two independent raters assessed each file according to ICD-10 definitions and Diagnostic and Statistical Manual of Mental Disorders, Fourth Edition, Text Revision and Fifth Edition criteria. Raters also completed the Clinical Global Impression–Severity (CGI-S) and the Global Assessment of Functioning (GAF).

**Primary outcome measure** Per cent between-rater agreement and positive predictive value (PPV). Intraclass correlation coefficients for the CGI-S and the GAF.

**Results** Eighty-four hypochondriasis and 122 dysmorphophobia files were received and analysed. The inter-rater agreement rate regarding the presence or absence of a diagnosis was 95.2% for hypochondriasis and 92.6% for dysmorphophobia. Sixty-seven hypochondriasis files (79.8%) and 111 dysmorphophobia files (91.0%) were considered 'true positive' cases (PPV=0.80 and PPV=0.91, respectively). CGI-S scores indicated that symptoms were moderately to markedly severe, while GAF scores suggested moderate impairment for hypochondriasis cases and moderate to serious impairment for dysmorphophobia cases. CGI-S and GAF inter-rater agreement were good for hypochondriasis and moderate for dysmorphophobia.

**Conclusions** The Swedish ICD-10 codes for hypochondriasis and dysmorphophobia are sufficiently

### Strengths and limitations of this study

► Randomly drawn sample of hypochondriasis and dysmorphophobia cases from all over Sweden.
► Thorough review of medical files by at least two independent expert raters.
► Good inter-rater reliability.
► No control diagnostic group.
► Limited number of cases and potential risk of selection bias.

valid and reliable for register-based studies. The results of such studies should be interpreted in the context of a possible over-representation of severe and highly impaired cases in the register, particularly for dysmorphophobia.

## INTRODUCTION

The National Board of Health and Welfare is a Swedish governmental agency that in 1964 established the National Patient Register (NPR), a health register with individual-level reporting of clinical diagnoses, which plays a crucial role in Swedish register-based epidemiological research.[1] The quality of the research conducted on NPR data is highly dependent on the diagnostic validity of the diagnostic codes.[2] Diagnoses in the NPR are coded according to the Swedish International Classification of Diseases (ICD) system, which was adapted from the WHO ICD classification system.[1] The validity of a wide range of ICD diagnostic codes in the Swedish NPR differs between diagnoses but is generally high.[1] Several diagnostic codes for psychiatric disorders have been examined and generally shown to be sufficiently valid and reliable for research purposes.[3–8]

Hypochondriasis (also known as illness anxiety disorder) and dysmophophobia (also known as body dysmorphic disorder) are two chronic and often severe psychiatric disorders associated with significant suffering and a high level of functional impairment.[9] Their estimated prevalence is 1%–2% for hypochondriasis[10] and around 2% for dysmorphophobia.[11 12] In the international version of the ICD-10,[13] hypochondriasis and dysmorphophobia are classed as somatoform disorders and share the same diagnostic code (F45.2). It is, therefore, not possible to separate the two disorders for clinical or research purposes.[13] By contrast, the Swedish version of the ICD-10 includes an additional code that allows clinicians to separately diagnose these two disorders. Specifically, hypochondriasis is coded F45.2, whereas dysmorphophobia is coded F45.2A.[14] This distinction is in line with the most recent classification of these disorders in the ICD-11, which considers them as two separate, but closely related diagnoses within the obsessive-compulsive spectrum.[9 15–18] Similarly, in the Diagnostic and Statistical Manual of Mental Disorders, Fifth Edition (DSM-5), hypochondriasis (illness anxiety disorder) and dysmorphophobia (body dysmorphic disorder) are two different disorders, although they appear under separate chapters (somatoform and obsessive-compulsive and related disorders, respectively).[9 19 20] Thus, the Swedish 'anomaly' in the ICD-10 potentially offers a unique opportunity for register-based studies on these disabling psychiatric conditions. However, the validity and reliability of these diagnostic codes have not yet been established.

This study employed a chart review methodology to establish the validity of the Swedish ICD-10 codes for hypochondriasis and dysmorphophobia in the NPR with the aim to assess whether these codes are suitable for future register-based studies.

## METHODS
### Procedures
After receiving approval from the regional ethical review board in Stockholm (2016/2399-31/5 and 2017/325-32), we requested 600 randomly selected personal identification numbers of individuals who ever received a diagnosis of either hypochondriasis (n=300) or dysmorphophobia (n=300) from the Swedish National Board of Health and Welfare. In accordance with the protocol approved by the ethical review board, individual patients were not asked for consent, as this would introduce selection biases. No weighting or other adjustments were done to randomly select the cases.

For hypochondriasis, we requested 300 files with the ICD-10 code F45.2 and all its subcodes (except for F45.2A), namely: F45.2B for nosophobia, F45.2C for cancer phobia, F452D for venerophobia and F45.2X for hypochondriasis, unspecified. For dysmorphophobia, we requested 300 files with the ICD-10 code F45.2A. The dates of registered diagnosis spanned from 1998 to 2016 for those with diagnoses assigned in inpatient clinics and

from 2001 to 2016 for those with diagnoses assigned in outpatient clinics. To be eligible for inclusion, a single ICD-10 diagnosis of hypochondriasis or dysmorphophobia at any time during this time period was sufficient, regardless of whether the diagnosis was primary or secondary or whether other comorbidities were present.

Following the procedures previously used in other validation studies by the research group,[3 7] once the random cases had been identified, we sent written requests to the corresponding archives or clinics, based on the hospital and medical specialty codes associated with the cases. Cases were excluded when we could not find the associated clinic (eg, the clinic was no longer operative), when the clinic did not reply or declined participation, when the diagnostic code under study was not documented in the received file or when there was no enough information in the received file to make a diagnostic judgement (eg, a description of clinical symptoms was not available). Figure 1 shows the flow for the inclusion of cases for each diagnosis. In total, we received 84 valid cases of hypochondriasis (including 72 cases diagnosed F45.2, hypochondriasis; one case diagnosed F45.2C, cancer phobia; 10 cases diagnosed F45.2X hypochondriasis, unspecified and one case diagnosed with both F45.2 and F45.2X) and 122 valid cases of dysmorphophobia available for analyses. The length of the received medical records ranged from 1 to about 1000 pages.

### Chart review
Two raters conducted an independent chart review of each medical record using a predefined scoring sheet (online supplemental material). A diagnosis was established independently by each of the two raters, based on all available information in the medical records. The raters for the hypochondriasis files were four clinical psychologists and one psychiatrist, with three of these five raters having a PhD degree. The raters of the dysmorphophobia files were six clinical psychologists and one psychiatrist, three of whom had a PhD. All had extensive clinical experience in the assessment and treatment of their respective disorders.

On revision of the chart, raters decided whether the ICD-10 definition of hypochondriasis or dysmorphophobia was met. Since the ICD-10 contains a narrative description of the disorder, rather than specific operational diagnostic criteria, raters were also asked whether the case under evaluation met diagnostic criteria for the corresponding diagnoses of hypochondriasis or body dysmorphic disorder according to the DSM, Fourth Edition, Text Revision (DSM-IV-TR) and, given the recent updates in the diagnostic criteria, also for the DSM-5 illness anxiety disorder or body dysmorphic disorder. If the two independent raters disagreed regarding the presence of a diagnosis, a third blind rater was asked to read the file. In a validation study, the expert rater is considered to be the gold standard and the diagnostic code in the file is the test. Hence, when a rater agreed with the diagnostic code in the file, the case was considered to be

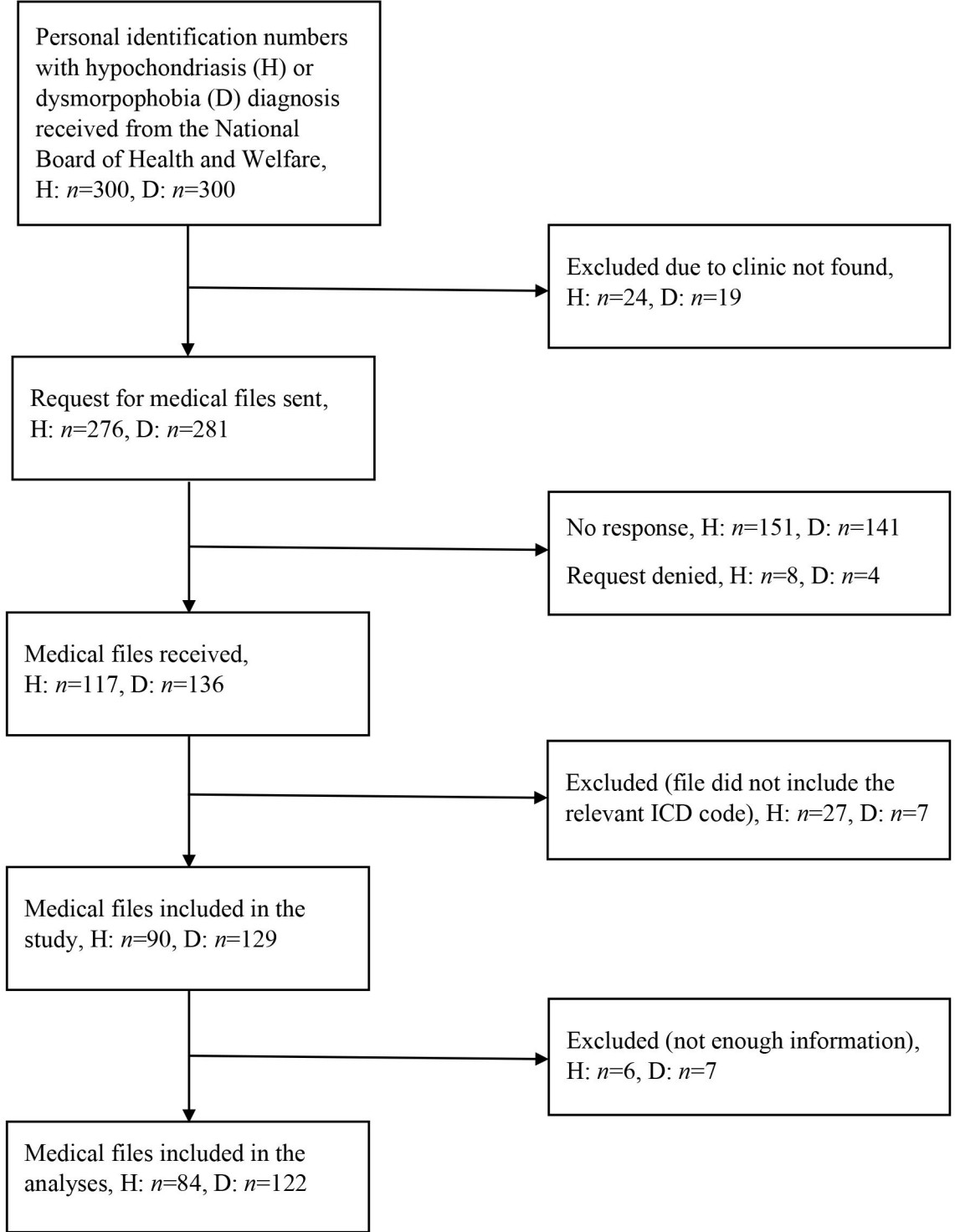

**Figure 1** Flowchart of requested and received patient files containing a hypochondriasis (H) or a dysmorphophobia (D) diagnosis code. ICD, International Classification of Diseases.

a 'true positive', and when a rater considered that a case did not meet criteria for the disorder in question, the case was defined as a 'false positive'. For false-positive cases, the raters were asked to provide the most likely alternative diagnosis, according to their clinical judgement.

Since the NPR only includes cases from specialist settings, raters also assessed symptom severity and global functioning related to hypochondriasis or dysmorphophobia in order to better evaluate the representativeness of the cases. These variables were assessed, respectively, by means of the Clinical Global Impression–Severity (CGI-S)[21 22] and the Global Assessment of Functioning (GAF)[23] rating scales. The CGI-S is a one-item measure assessing the severity of psychopathology from 1 to 7, where 1 is 'normal' and 7 is 'among the most extremely ill patients.'[21] The CGI-S has shown good

internal consistency and concurrent validity.[24] The GAF is also a one-item measure (ranging from 1 to 100) used in psychiatry to assess the general social, occupational and psychological functioning of adults.[25] Scores in the 1–10 range indicate a severely impaired functioning with persistent danger for self or others, whereas scores in the 91–100 range indicate superior functioning with no symptoms. The GAF has shown good validity and reliability in the assessment of global functioning in psychiatric patients.[25] Both the CGI-S and the GAF are generally rated in reference to the time of the assessment. Because of the nature of this study, raters were instead asked to make an estimation of the average severity and function of the patient for the whole time covered in the file.

### Statistical analyses

The rate of agreement between the two evaluators of each file was calculated. Since the raters' responses in both hypochondriasis and dysmorphophobia cases were very imbalanced (ie, the answer 'yes' indicating the presence of the disorder was much more common for all raters, compared with 'no'), we did not use Kappa statistics to examine inter-rater reliability. This was because in cases with this kind of imbalance in responses, Kappa results may be misleading, showing a paradox where the coefficients are low despite high agreement rates.[26 27] Instead, we calculated the per cent agreement between the two initial raters, which is a valid alternative to Kappa coefficients when using well-trained raters who are not likely to guess.[28] The per cent agreement is the per cent of ratings where both raters made the same judgement.

Furthermore, for each diagnosis, we calculated the PPV and their corresponding 95% CIs. The PPV is calculated by dividing the cases diagnosed correctly by the sum of the true positives and the false positives.

To assess the inter-rater agreement for the CGI-S and the GAF scales, intraclass correlation coefficients (ICC) with 95% CIs were calculated based on one-way mixed-effects model for average measures, absolute agreement.[29] Stata V.15.1 (StataCorp LLC) was used for all the analyses.

### Patient and public involvement

No patients were involved in setting the research question nor were they involved in developing plans for the study design or data analysis. There are no plans to directly disseminate the results of the research to study participants or the relevant patient community. The dissemination to the public will be achieved through media outreach (eg, press release and communication) on publication of this study.

### RESULTS

### Validity and reliability of hypochondriasis codes in the NPR

A total of 84 cases with a register diagnosis of hypochondriasis (45 women, 53.6%) were included in the analysis. The cases came mostly from psychiatric clinics (n=75, 89.3%), followed by emergency units (n=3,

3.6%), internal medicine clinics (n=2, 2.4%), neurology clinics (n=2, 2.4%), gynaecology clinics (n=1, 1.2%) and oncology clinics (n=1, 1.2%).

In 80 (95.2%) of the 84 cases, the initial two raters agreed on the presence or absence of a hypochondriasis diagnosis in the file. A third independent rater reviewed the files of four cases where there was a disagreement between the two initial raters: two of these four files were considered true positives and another two were considered false positives.

In total, 67 (79.8%) of the 84 cases were defined as true positives. In the majority of cases (n=63, 94.0%), both raters considered that the criteria were met according to all three diagnostic systems (ie, ICD-10, DSM-IV-TR and DSM-5). In the remaining four cases, raters considered that the ICD-10 definition and the DSM-IV-TR diagnostic criteria were met, but not the DSM-5 criteria.

The 67 true positive cases translated into a PPV of 0.80 (95% CI 0.70 to 0.88). For the remaining 17 false-positive cases, the most frequent alternative diagnosis was dysmorphophobia (n=11), followed by psychotic disorder (n=4), borderline personality disorder (n=2), major depressive disorder (n=2), somatisation disorder (n=2), somatoform disorder, unspecified (n=2) and obsessive-compulsive disorder (n=2) (table 1). Of note, eight of the nine cases from non-psychiatric clinics were considered to be true positives (ie, correctly classified).

### Validity and reliability of dysmorphophobia codes in the NPR

A total of 122 cases with a register diagnosis of dysmorphophobia (83 females, 68.0%) were included in the analysis. The majority of files (n=106, 86.9%) came from psychiatric clinics, with the remaining coming from dermatology clinics (n=11, 9.0%), plastic surgery clinics (n=4, 3.3%) and one from a gynaecological clinic (n=1, 0.8%).

There was an agreement between the two initial raters regarding the presence or absence of a dysmorphophobia diagnosis in 113 of the 122 files (92.6%). Of the nine cases where there was a disagreement, the third independent rater concluded that two were true positives and seven were false positives.

In total, 111 (91.0%) of the 122 cases were classed as true positives. In the vast majority of cases (n=108, 97.3%), the criteria were met according to all three diagnostic systems, according to both raters. In the three remaining cases, raters considered that the ICD-definition was met, but not all criteria according to the more stringent diagnostic systems, DSM-IV-TR and DSM-5.

Based on the 111 true-positive cases, the PPV was 0.91 (95% CI 0.84 to 0.95). For the remaining 11 cases defined as false positives, the most frequent alternative diagnoses were excoriation (skin-picking) disorder (n=3), factitial dermatitis (n=3), eating disorder (n=3), hypochondriasis (n=2), pervasive development disorder (n=2) and psychotic disorder (n=2) (table 1). Of note, the four cases from plastic surgery clinics were considered to be true positives, as were 7 of the 11 cases (63.6%) from

**Table 1** Alternative diagnoses for false positive cases of hypochondriasis (n=17) and dysmorphophobia (n=11).

| Hypochondriasis | n | Dysmorphophobia | n |
|---|---|---|---|
| Dysmorphophobia | 11 | Excoriation (skin-picking) disorder | 3 |
| Psychotic disorder | 4 | Factitial dermatitis | 3 |
| Somatisation disorder | 2 | Eating disorder, unspecified | 3 |
| Somatoform disorder, unspecified | 2 | Hypochondriasis | 2 |
| Obsessive-compulsive disorder | 2 | Psychotic disorder | 2 |
| Major depressive disorder | 2 | Pervasive developmental disorder | 2 |
| Borderline personality disorder | 2 | Delusional disorder | 1 |
| General anxiety disorder | 1 | Somatisation disorder | 1 |
| Anxiety disorder, unspecified | 1 | Obsessive-compulsive disorder | 1 |
| Bipolar disorder | 1 | Trichotillomania | 1 |
| Pervasive developmental disorder | 1 | Social phobia | 1 |
| Substance dependence disorder | 1 | Generalised anxiety disorder | 1 |
| Acute stress reaction | 1 | Post-traumatic stress disorder | 1 |
| | | Gender identity disorder | 1 |
| | | Borderline personality disorder | 1 |

Numbers do not add up to the total of false positive cases (n=17 for hypochondriasis and n=11 for dysmorphophobia) given that, for multiple cases, raters suggested more than one alternative diagnosis.

dermatology clinics, while the one case from a gynaecological clinic was considered to be a false positive.

### Severity and global function

CGI-S and the GAF data were available for 63 of the 67 true-positive hypochondriasis cases; in the remaining four cases, raters had not scored the scales due to lack of information in the medical file, thus the information was missing. The mean score for the CGI-S was 4.49 (SD=1.01, median=5, IQR=1) for rater 1 and 4.57 (SD=0.73, median=5, IQR=1) for rater 2, indicating moderate to marked severity of the assessed cases (figure 2, panel A). The inter-rater reliability for the CGI-S was good (ICC=0.75 (95% CI 0.59 to 0.85)). The mean GAF score was 54.40 (SD=9.41, median=50, IQR=12) for rater 1 and 52.63 (SD=9.45, median=49, IQR=15) for rater 2, indicating moderate impairment of global functioning (figure 2, panel A). The inter-rater reliability for the GAF was also good (ICC=0.81 (95% CI 0.69 to 0.89)).

For dysmorphophobia, CGI-S and GAF scores were available for 94 of the 111 true-positive cases; in the remaining 17 cases, raters had not scored the scales due to lack of information in the medical file. The mean score for the CGI-S was 4.70 (SD=1.20, median=4, IQR=2) for rater 1 and 4.99 (SD=0.71, median=5, IQR=1) for rater 2, indicating moderate to marked severity of the assessed cases (figure 2, panel B). The inter-rater reliability for the CGI-S was moderate (ICC=0.61 (95% CI 0.41 to 0.74)). The mean GAF-score was 47.98 (SD=12.77, median=52.5, IQR=15) as assessed by rater 1 and 47.79 (SD=7.32, median=51, IQR=8) as assessed by rater 2, indicating serious impairment in global functioning (figure 2, panel

B). The inter-rater reliability for the GAF was moderate (ICC=0.65 (CI 0.48 to 0.77)).

### DISCUSSION

This study evaluated the validity and reliability of the diagnostic codes for hypochondriasis and dysmorphophobia in the Swedish NPR using a chart review design, which is considered to be the gold-standard procedure for assessing diagnostic validity.[3] Our results showed that the diagnostic validity of both disorders is generally good, with a PPV of 0.80 for hypochondriasis and 0.91 for dysmorphophobia. These findings are in line with those of previous studies validating other psychiatric disorders in the NPR, including bipolar disorder (PPV=0.81–0.91),[4] schizophrenia (PPV=0.91–1.0),[8] obsessive-compulsive disorder (PPV=0.55–0.96),[3] chronic tic disorders (PPV=0.86–0.97)[3] and social anxiety disorder (PPV=0.72–0.88).[7] Furthermore, the inter-rater agreement for both hypochondriasis and dysmorphophobia was satisfactory.[30]

Nonetheless, 20% of the hypochondriasis files and almost 10% of the dysmorphophobia files were misclassified. For the majority of the misclassified hypochondriasis files (64.7%), dysmorphophobia was suggested as the most likely alternative diagnosis. Since both disorders share the same diagnostic code, it is likely that at least a proportion of those cases were a result of coding errors (ie, the clinician not knowing that the F45.2A was the corresponding code for dysmorphophobia). In the same way, a smaller but non-negligible proportion of dysmorphophobia cases (18.2%) was judged to better correspond to a diagnosis of hypochondriasis. Thus, it seems that the high proximity

**A**

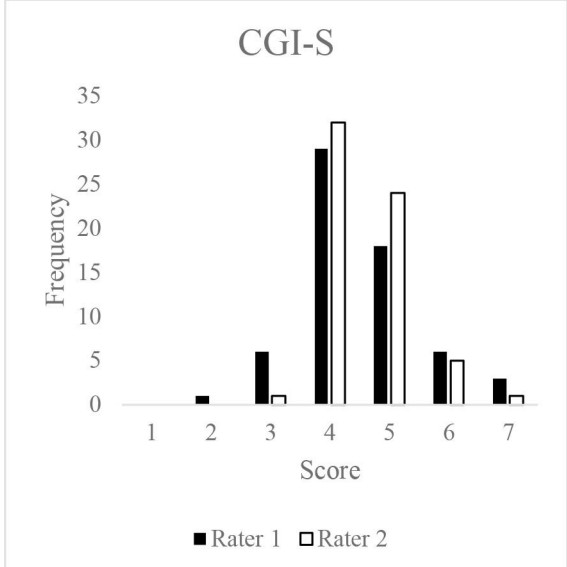
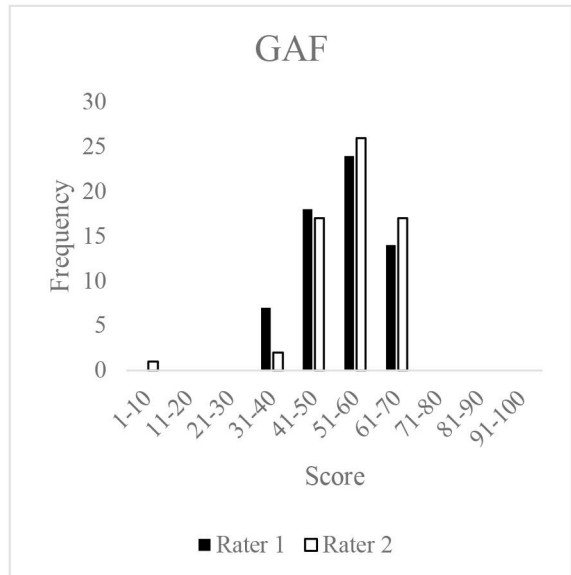

**B**

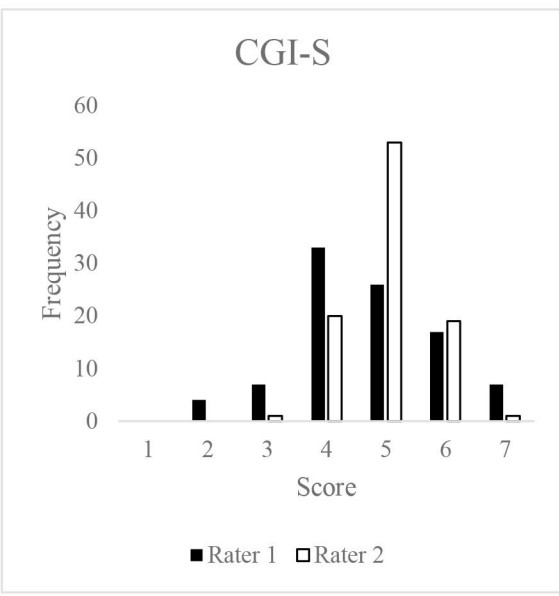
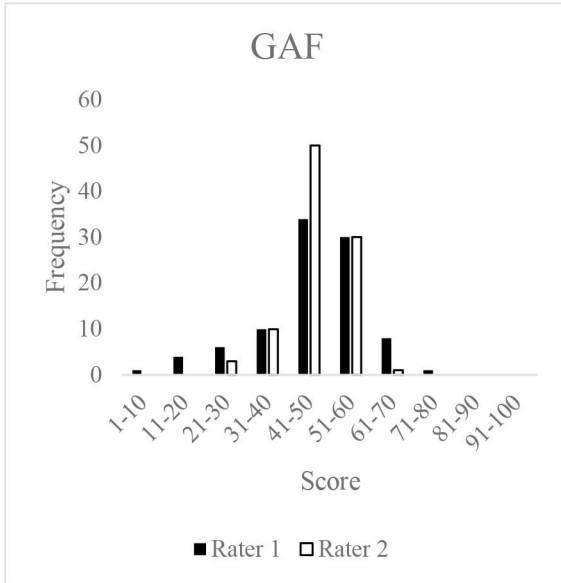

**Figure 2** Score distribution of the Clinical Global Impression—Severity (CGI-S) and Global Assessment of Functioning (GAF) by rater, for hypochondriasis (A) and for dysmorphophobia (B).

and similarity of these adjacent codes poses a challenge for clinicians and may have implications for register-based studies. Because patients receive a new diagnostic code with every specialist visit, individuals in the registers often receive multiple diagnostic codes over time; in this context, it may be wise to question the validity of cases receiving both diagnoses during the follow-up. For this reason, we suggest that future register-based studies using the ICD-10 diagnosis of hypochondriasis (F45.2) should exclude individuals with recorded dysmorphophobia codes (F45.2A) at any point during the follow-up, and *vice versa*, in order to reduce the risk of potential misclassification to a minimum.

An additional issue in the register-based epidemiological studies conducted in Sweden is that the NPR only includes diagnoses assigned by physicians in specialist care settings. Furthermore, it is well known that individuals with hypochondriasis and dysmorphophobia are often reluctant to seek mental health support due to embarrassment about symptoms, poor insight and a preference for non-psychiatric care (eg, cosmetic procedures in dysmorphophobia, somatic care in hypochondriasis).[31–34]

As a result, both conditions are presumably severely underdiagnosed[10 12] and it could be assumed that the patients in the registers are more severe and less functional than the average patient. This may affect the generalisability of the results from register-based studies to non-specialist clinical settings. Nonetheless, the hypochondriasis sample had a broad distribution of severity and global functioning scores, with most patients being moderately ill and having a moderately impaired function. Regarding the dysmorphophobia files, distributions of the severity and functioning variables were somewhat skewed to the more severe end of the spectrum.

The main strengths of this study are the random selection of cases from all over Sweden and the thorough review of the medical files by two or three independent expert raters, showing good inter-rater agreement. However, there are also some limitations to consider. First, the study had no control diagnostic group, which may result in an increased risk of overconfirming the target diagnosis. Second, there is a risk for selection bias, given that only 28% of the requested hypochondriasis files and 41% of the requested dysmorphophobia files could be included in the final analyses. However, because the reasons for not including the files were mostly practical (eg, some clinics did no longer exist, had confidentiality concerns or no personnel available to send the files), we assume that a systematic bias is unlikely. Third, we were unable to evaluate the validity of the hypochondriasis subtypes separately given the small number of files received containing these specific codes (11 files containing only the codes F45.2C or F45.2X). Finally, since the raters did not interview the patients in person, the scoring of the CGI-S and the GAF rating scales should be seen as a general clinical estimate of the patients' severity and general function, rather than a precise assessment.

## Conclusions

The ICD-10 codes for both hypochondriasis and dysmorphophobia in the Swedish NPR are sufficiently valid and reliable for their use in register-based studies. However, the results of such studies should be interpreted in the context of a possible over-representation of severe and highly impaired cases in the register, particularly for dysmorphophobia.

**Contributors** LFC, DM-C, ES, CR and JH were involved in the conception of the research question and designed the study protocol. AV-P and SK were the data managers and administrators for the project. DR, MG, VZI, JL, SÖ, OF, JI and KI were independent raters in the chart review. DR contributed to the data management and performed the statistical analyses. DR and LFC drafted the manuscript. LFC and DM-C provided supervision. All authors contributed to the final version of the manuscript by providing substantial intellectual contributions. The authors read and approved the final manuscript. LFC is responsible for the overall content of the manuscript as guarantor.

**Funding** This study was supported by a grant from Region Stockholm (ALF Medicin grant reference number 20180078) awarded to LFC. AVP was supported by a fellowship from the Alicia Koplowitz Foundation (award/grant number N/A).

**Competing interests** DM-C receives royalties for contributing articles to UpToDate, Wolters Kluwer Health and Elsevier. LFC receives royalties for contributing articles to UpToDate, Wolters Kluwer Health. The rest of authors declare that they have no competing interests.

**Patient consent for publication** Not applicable.

**Ethics approval** Regional ethical review board in Stockholm (2016/2399-31/5 and 2017/325-32).

**Provenance and peer review** Not commissioned; externally peer reviewed.

**Data availability statement** No data are available.

**ORCID iDs**
Daniel Rautio http://orcid.org/0000-0002-8657-8481
Alba Vilaplana-Pérez http://orcid.org/0000-0002-9501-8160
Oskar Flygare http://orcid.org/0000-0002-2017-3940
Christian Rück http://orcid.org/0000-0002-8742-0168
David Mataix-Cols http://orcid.org/0000-0002-4545-0924
Lorena Fernández de la Cruz http://orcid.org/0000-0002-1571-5485

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
