## [Reviewer comments · BMJ Open]

ARTICLE DETAILS

TITLE (PROVISIONAL)	Validity and reliability of the diagnostic codes for hypochondriasis and dysmorphophobia in the Swedish National Patient Register: A retrospective chart review
AUTHORS	Rautio, Daniel; Vilaplana-Pérez, Alba; Gumpert, Martina; Ivanov, Volen; Linde, Johanna; Österman, Susanna; Flygare, Oskar; Isung, Josef; Isumura, Kayoko; Krig, Sonja; Serlachius, Eva; Högström, Jens; Rück, Christian; Mataix-Cols, David; Fernández de la Cruz, Lorena

VERSION 1 – REVIEW

REVIEWER	Stewart, Robert King's College London, Institute of Psychiatry
REVIEW RETURNED	11-Jun-2021

GENERAL COMMENTS	This manuscript reports a straightforward study in which the authors investigated positive predictive values for hypochondriasis and dysmorphophobia in the Swedish National Patient Register against an expert case note review gold standard. There are limitations in the approach used (particularly that the experts presumably knew that all cases had been assigned one of these diagnoses, and I assume could see the assigned diagnosis in the case notes reviewed); however, the authors acknowledge this. In addition, the study was not seeking to investigate how many people might have these health conditions and not be present on the Register; however, I can see that this would involve another study design entirely. Therefore personally I feel that the paper, which is clearly written, presents the findings appropriately and I have no recommended changes to make.
--

REVIEWER	Hartmann, Andrea Universität Konstanz, Psychology
REVIEW RETURNED	11-Oct-2021

GENERAL COMMENTS	The manuscript describes a chart-review study with the aim to assess validity and reliability of ICD-10 codes of illness anxiety disorder and body dysmorphic disorder in the Swedish National Patient Register. The Swedish Patient Register has produced many datasets for a great number of excellent studies answering highly relevant questions without putting any additional burden on single patients. With the subcodes for the two disorders being different, it would be a great opportunity to investigate research questions in groups of patients of either disorder, while being able to trust the reliability of the diagnoses. This indeed provides a unique opportunity as other systems in other countries (e.g., data from insurance companies in Germany) do not differentiate codes from
--

	these disorders. The paper is well written and description of methods are clear. I only have a few very minor comments:  - Is there any difference between primary and secondary diagnoses in the register and were only those datasets with primary diagnoses of either disorder extracted? What about comorbidity (between the disorders and with others?) - I like the practical recommendation on pages 13/14 with regard to inclusion of data sets with both diagnoses included over time. - While I agree with the authors with regard to the statement on page 14 lines 32-52, I think that the paragraph does not clearly fit here. They might want to consider removing it or linking it to the previous and latter ones.
--	---

VERSION 1 – AUTHOR RESPONSE

Reviewer 1

Dr. Robert Stewart, King's College London

Comments to the Author:

This manuscript reports a straightforward study in which the authors investigated positive predictive values for hypochondriasis and dysmorphophobia in the Swedish National Patient Register against an expert case note review gold standard. There are limitations in the approach used (particularly that the experts presumably knew that all cases had been assigned one of these diagnoses, and I assume could see the assigned diagnosis in the case notes reviewed); however, the authors acknowledge this. In addition, the study was not seeking to investigate how many people might have these health conditions and not be present on the Register; however, I can see that this would involve another study design entirely. Therefore personally I feel that the paper, which is clearly written, presents the findings appropriately and I have no recommended changes to make.

We thank the Reviewer for the positive feedback on our manuscript.

Reviewer 2

Dr. Andrea Hartmann, Universität Konstanz

Comments to the Author:

The paper is well written and description of methods are clear. I only have a few very minor comments:

- Is there any difference between primary and secondary diagnoses in the register and were only those datasets with primary diagnoses of either disorder extracted? What about comorbidity (between the disorders and with others?)

We thank the Reviewer for the generally positive appraisal of our manuscript.

In response to their query, we thank the opportunity to clarify that, despite the Swedish National Patient Register includes primary and secondary diagnosis, this study did not make a differentiation and both categories were extracted for the chart review. In the same way, the diagnoses of the disorders under study were included regardless of comorbidities. We have now clarified this in the Methods, Procedures section (page 6): "To be eligible for inclusion, a single ICD-10 diagnosis of hypochondriasis or dysmorphophobia at any time during this time period was sufficient, regardless of whether the diagnosis was primary or secondary or whether other comorbidities were present."

- I like the practical recommendation on pages 13/14 with regard to inclusion of data sets with both diagnoses included over time.

Thank you.

- While I agree with the authors with regard to the statement on page 14 lines 32-52, I think that the paragraph does not clearly fit here. They might want to consider removing it or linking it to the previous and latter ones.

Following the Reviewer's recommendation, we have now significantly shortened this paragraph and linked it with the previous.

VERSION 2 – REVIEW

REVIEWER	Hartmann, Andrea Universität Konstanz, Psychology
REVIEW RETURNED	02-Nov-2021
GENERAL COMMENTS	The authors have been responsive to both reviewers' minor comments. I do not have any additional/new concerns.